# Virus-Associated Biomarkers in Oropharyngeal and Nasopharyngeal Cancers and Recurrent Respiratory Papillomatosis

**DOI:** 10.3390/microorganisms9061150

**Published:** 2021-05-27

**Authors:** Shigeyuki Murono

**Affiliations:** Department of Otolaryngology, Fukushima Medical University, Fukushima 960-1295, Japan; murono@fmu.ac.jp; Tel.: +81-24-547-1321

**Keywords:** human papillomavirus, oropharyngeal cancer, DNA, plasma, oral rinse, biomarker, recurrent respiratory papillomatosis

## Abstract

Nasopharyngeal cancer (NPC) is known to be associated with Epstein–Barr virus (EBV). Pre-treatment and post-treatment detection of plasma cell-free EBV DNA has been shown to be useful as a diagnostic as well as a prognostic factor in NPC. On the other hand, the incidence of human papillomavirus (HPV)-associated oropharyngeal cancer (OPC) is increasing. In contrast to cervical cancer, which is classically known to be an HPV-associated malignancy, HPV testing is not clinically applied for OPC, except for p16 immunostaining as a surrogate marker of HPV infection. One of the major characteristics of HPV-associated OPC is its association with a good prognosis compared with non-HPV-associated OPC. However, some patients still have a poor prognosis. Another characteristic of HPV-associated OPC is the distinct risk factor of high sexual activity. Establishing a biomarker for the prediction of the prognosis before and/or after initial treatment, as well as for diagnosis in populations at high risk, is of marked interest. With this background, HPV DNA detection in plasma and oral rinses has become an area of focus. In this review, the current significance of HPV DNA detection in plasma and oral rinse samples, as well as serum HPV antibody levels, is evaluated.

## 1. The First Human Virus-Associated Cancer: Epstein–Barr Virus and Nasopharyngeal Cancer

### 1.1. Plasma Cell-Free Epstein–Barr Virus DNA in Nasopharyngeal Cancer Patients

Classically, the most well-known virus-associated head and neck cancer is nasopharyngeal cancer (NPC), which is closely associated with Epstein–Barr virus (EBV). In situ hybridization for EBV-encoded RNA (EBER) can clearly identify EBV-infected tumor cells, whereas elevation of the serum EBV titer has been useful as a diagnostic adjunct [1,2]. In addition, quantitation of plasma/serum cell-free EBV DNA has had an impact as a biomarker for NPC diagnosis and prognosis after treatment [3]. Lo et al. showed for the first time that cell-free EBV DNA could be detected in 55 (96%) of 57 patients with NPC [3]. In addition, patients with undetectable post-treatment plasma EBV DNA showed no clinically residual tumors, whereas patients with detectable post-treatment plasma EBV DNA showed residual or recurrent tumors [3]. A meta-analysis of 15 published papers revealed that the sensitivity and specificity of plasma EBV DNA were 91% and 93%, respectively, which were higher than those of serum EBV DNA, at 84% and 76%, respectively [4]. With respect to prognosis, a meta-analysis of 14 published papers revealed that both patients with high plasma EBV DNA before treatment and those with persistent plasma EBV DNA after treatment showed significantly poorer overall survival (hazard ratio (HR), 2.81; 95% CI, 2.44–3.24; *p* < 0.00001, and HR, 4.26; 95% CI, 3.26–5.57; *p* < 0.00001, respectively) [5]. Wang et al. reported that a higher pre-treatment plasma EBV DNA level was associated with poorer overall survival (*p* = 0.0295) and relapse-free survival (*p* = 0.0163) [6]. Furthermore, the residual post-treatment plasma EBV level was closely associated with poorer overall survival (*p* < 0.0001) and relapse-free survival (*p* < 0.0001) [7]. Based on these findings, clinical studies investigating the significance of adjuvant chemotherapy in patients with residual plasma EBV DNA are ongoing.

### 1.2. Other EBV-Associated Malignant Tumor

Extranodal natural killer/T-cell lymphoma, nasal type (ENKL) is another well-known EBV-associated head and neck malignancy. Lei et al. first reported that plasma EBV DNA was a useful tumor marker for the diagnosis, disease monitoring, and prediction of outcomes in patients with ENKL [8]. Suzuki et al. showed that pre-treatment plasma EBV DNA was detected in 14 (44%) of 32 patients with ENKL [9]. On the other hand, Wang et al. reported that pre-treatment plasma EBV DNA was detected in 58 (84%) of 69 stage I and stage II ENKL patients, which was higher than the results of previous studies [10]. Suzuki et al. also showed better 3-year overall survival in patients who were negative for pre-treatment plasma EBV DNA than those who were positive for it [9]. Similarly, patients with low pre-treatment plasma EBV DNA showed better 3-year overall survival and 3-year progression-free survival than those with high pre-treatment plasma EBV DNA (97% vs. 66% and 79% vs. 52%, respectively) [10]. Furthermore, patients who were positive for post-treatment plasma EBV-DNA showed poorer 3-year overall survival and 3-year progression-free survival than those who were negative for post-treatment plasma EBV DNA (92% vs. 70% and 78% vs. 51%, respectively) [10]. These results indicate that plasma EBV DNA can be a biomarker not only of NPC but also of ENKL.

## 2. Human Papillomavirus as a Causative Agent Not Only for Cervical Cancer, but also for Oropharyngeal Cancer

Head and neck cancer has typically been diagnosed in older patients, in association with the heavy use of tobacco and alcohol [11]. On the other hand, it has become apparent that human papillomavirus (HPV) is a principal cause of a distinct form of oropharyngeal cancer (OPC) [12]. The incidence of head and neck cancer is slowly declining globally, in part because of decreased use of tobacco [11,13]. Conversely, cases of HPV-associated OPC are increasing, predominantly among younger people in North America and northern Europe—Chaturvedi et al. showed that the HPV prevalence in OPC patients significantly increased from 16.3% in the 1980s to more than 72.7% in the 2000s in the United States [11,14].

With an increasing incidence of HPV-positive OPC, the projected annual number of OPC patients has surpassed that of cervical cancer patients since 2010 [6,14]. On the other hand, HPV has been recognized as a causative agent of cervical cancer. Currently, this strong association is clinically applied to two very effective prevention strategies for cervical cancer: cervical screening with primary HPV testing and vaccination against HPV [15]. In 2020, it was proven for the first time that quadrivalent HPV vaccination was associated with a substantially reduced risk of invasive cervical cancer at the population level among Swedish girls and women aged from 10 to 30 years old [16]. On the other hand, the effectiveness of prophylactic HPV vaccination is less well defined for OPC than for anogenital and cervical cancers [11].

From the perspective of cervical cancer screening, cytology-based screening programs have been implemented in many countries. On the other hand, various HPV-based screening methods—including HPV DNA tests by direct genomic detection or by amplification of a viral DNA fragment using polymerase chain reaction, and HPV mRNA tests to detect the expression of *E6* and *E7* viral oncogenes—have been found to have higher sensitivity than cytology but somewhat lower specificity in cervical cancer [17]. Therefore, co-testing with cytology and HPV, and lately HPV testing alone, have been proposed for cervical cancer screening [17]. Furthermore, self-sampling for HPV testing has been developed [17]. However, no approved HPV-based screening, except for p16 immunostaining as a surrogate marker of HPV infection in histopathological diagnosis, is available for HPV-positive OPC. In this review, we focus on viral DNA in virus-associated head and neck cancer, and summarize the usefulness of HPV DNA detection in oral rinse samples, as well as in plasma, as a diagnostic and prognostic marker of HPV-associated OPC.

## 3. Plasma Cell-Free HPV DNA in Oropharyngeal Cancer Patients

### 3.1. Oropharyngeal Cancer

The question of whether plasma/serum cell-free HPV DNA is useful as a biomarker of HPV-associated OPC, like EBV DNA in NPC, is of marked interest. Capone et al. first demonstrated the detection and quantitation of HPV DNA in the sera of four out of nine HPV-associated OPC patients [18]. However, most reports were published in the 2010s, as reviewed by Jensen et al. in 2018 [19]. Although the pre-treatment sensitivity was around 65% in earlier reports, it has reached 90% in recent reports (Table 1) [20,21,22,23,24]. In addition, some reports have demonstrated the quantitation of plasma HPV DNA viral load, which may present clinical relevance (Table 2) [20,22,24].

Interestingly, plasma HPV DNA reflects the tumor status, as observed in NPC, with Cao et al. demonstrating that serial measurements in 14 patients showed a rapid decline in HPV DNA, which became undetectable on the completion of radiotherapy, whereas the HPV DNA level increased to a detectable level at metastasis in three patients [20]. Similarly, Lee et al. reported that 36 patients who were negative for post-treatment HPV DNA showed no recurrence, whereas the remaining patient, who was positive for post-treatment HPV DNA, showed recurrence and metastasis [23]. These results suggest that post-treatment plasma HPV DNA correlates with the clinical response. However, pooled analyses from four studies demonstrated that the sensitivity and specificity of the residual detection of plasma HPV DNA for the recurrence of HPV-associated OPC were 54% and 98%, respectively [19]. On the other hand, the clearance of plasma HPV DNA has recently become an area of focus. Chera et al. reported that favorable clearance, defined as having a high baseline plasma HPV DNA copy number (>200 copies/mL) and >95% clearance of plasma HPV DNA by day 28 of chemoradiotherapy, led to better survival with no persistent or recurrent regional disease after the initial treatment [24].

### 3.2. Cervical Cancer

As mentioned before, cervical cancer is known to be associated with HPV. However, in contrast to OPC, there have been limited reports investigating the usefulness of plasma/serum HPV DNA in cervical cancer. In early reports, positivity was low—Pornthanakasem et al. showed that six (12%) out of 50 patients and Dong et al. showed 12 (7%) out of 175 patients tested positive [25,26]. Results have recently improved, with Jeannot et al. showing 61 (87%) of 70 patients and Cheung et al. showed 85 (62%) of 135 patients [27,28].

## 4. Pre-Treatment HPV DNA in Oral Rinses of Oropharyngeal Cancer Patients

As mentioned before, the association of HPV with cervical cancer is clinically applied to cervical screening with primary HPV testing. On the other hand, such a screening system has yet to be established for OPC, although the incidence of HPV-associated OPC has been increasing. From this perspective, oral rinses or saliva may be attractive samples due to the fact that their collection is non-invasive.

Smith et al. reported in 2004 that the sensitivity and specificity of HPV DNA detection in oral rinse samples were 58% and 85%, respectively, when combining 67 OPC and 126 oral cancer patients [29]. Several reports describing the detection of HPV DNA in oral rinses or saliva have been published since then. Some focused on OPC, whereas others focused on OPC and oral cancer or all head and neck cancers, as shown in Table 3 [21,29,30,31,32,33,34,35,36,37,38,39,40,41,42,43,44,45,46,47]. Gipson et al. reviewed seven studies, including two using swabs, and demonstrated that the combined sensitivity and specificity were 72% (95% CI, 45–89) and 92% (95% CI, 82–97), respectively, in 2018 [46]. Figure 1 presents previous reports, and shows that the specificity was 90% or higher in most studies, whereas the sensitivity varied among studies, with the highest being 93%.

As mentioned before, the quantitation of viral load may present clinical relevance in plasma. Similarly, some reports have investigated the HPV DNA viral load in oral rinse samples (Table 4) [32,36,44]. However, unlike studies of plasma, viral load seems to vary among these studies.

## 5. Post-Treatment HPV DNA in Oral Rinse of Oropharyngeal Cancer Patients

Residual plasma EBV DNA after the completion of initial treatment is associated with a poorer prognosis in NPC patients. Similarly, as mentioned before, residual plasma HPV DNA could be a predictive indicator of a poorer prognosis in HPV-associated OPC patients. From a similar perspective, the dynamics of HPV DNA in oral rinses after treatment has been of marked interest. Chuang et al. reported that four of 20 patients with positive pre-treatment HPV DNA ultimately developed recurrence, and two of these four patients had HPV16-positive post-treatment salivary rinses, showing 50% sensitivity [48]. On the other hand, among 16 HPV16-positive patients who did not show recurrence, none had HPV16-positive salivary rinses after treatment, showing 100% specificity.

Ahn et al. demonstrated that the sensitivity and specificity of pre-treatment HPV DNA detection in saliva in 83 OPC patients, including 72 HPV-positive and 11 HPV-negative patients, were 53% and 100%, respectively [21]. Although the sensitivity seems to be relatively low compared with data from a systematic review by Gipson, they reported that four out of 38 OPC patients who were positive for pre-treatment HPV16 DNA had persistently detectable HPV16 DNA in their saliva after treatment [21,46]. In a multivariable analysis, a post-treatment saliva HPV-positive status was associated with a higher risk of recurrence (HR, 10.7; 95% CI, 2.36–48.50) (*p* = 0.002). Overall survival was reduced among those with a post-treatment saliva HPV-positive status (HR, 25.9; 95% CI, 3.23–208.00; *p* = 0.002). Rettig et al. prospectively demonstrated that persistent HPV16 DNA was detected in oral rinse samples in five of 124 HPV-associated OPC patients, whereas it was detected in 67 of 124 patients in pre-treatment oral rinse samples [36]. Furthermore, patients with persistent HPV16 DNA after treatment showed significantly poorer progression-free survival (HR, 29.7; 95% CI, 9.0–98.2; *p* < 0.001) and overall survival (HR, 23.5; 95% CI, 4.7–116.9; *p* < 0.001) than patients without persistent HPV16 DNA. Although the prognosis was not stated, Yoshida et al. also demonstrated that persistent HPV DNA in oral rises was only detected in one of eight HPV-associated OPC patients without clinically residual disease after treatment [40]. These results suggest that detectable pre-treatment HPV DNA in oral rinses is cleared after treatment, and that persistent HPV DNA is associated with a poorer prognosis.

A larger-scale prospective study was recently published by Fakhry et al. [45]. Of their 396 patients—including 170 (seven HPV-positive and 163 HPV-negative) with oral cancer, 217 (187 HPV-positive and 30 HPV-negative) with OPC, and nine (eight HPV-positive and one HPV-negative) with unknown primary cancer—the sensitivity and specificity of pre-treatment HPV16 DNA in oral rinses were 81% and 100%, respectively. The detection of tumor-type oral HPV DNA among patients treated with primary radiotherapy with or without chemotherapy decreased from 85% at baseline to 9% after the completion of radiotherapy (*p* < 0.001). Interestingly, the tumor-type HPV DNA load decreased rapidly during primary radiotherapy for most patients (24% relative reduction per weekly visit; *p* < 0.001). On the other hand, the prevalence rates of tumor-type oral HPV DNA before and after primary surgical resection were 69% and 14%, respectively (*p* < 0.001). Furthermore, in the subset of patients who required adjuvant radiotherapy, the prevalence decreased from 70% to 38% after surgical resection and then decreased to 1% after radiotherapy (*p* < 0.001). Significantly lower 2-year overall survival (68% vs. 95%, respectively; adjusted HR, 6.61; 95% CI, 1.86–23.44; *p* = 0.003) and recurrence-free survival (55% vs. 88%, respectively; adjusted HR, 3.72; 95% CI, 1.71–8.09; *p* < 0.001) were observed among HPV-positive patients with persistent detection of tumor-type HPV in oral rinse samples after treatment than in those without detectable tumor-type HPV in oral rinse samples after treatment. These results suggest that HPV DNA in oral rinse samples rapidly decreases with treatment, and that the persistent detection of HPV DNA in oral rinse samples is associated with poorer survival and recurrence.

## 6. Serum HPV Antibodies in HPV-Associated Oropharyngeal Cancer Patients

Aside from HPV DNA detection, many investigators have also been interested in serum HPV antibodies in HPV-associated cancer, including OPC, as described in a review by Mirghani [49]. Dahlstrom et al. demonstrated significantly higher positive rates of NE2 (86.5% vs. 61.1%, respectively), E6 (79.2% vs. 44.4%, respectively), and any E (89.6% vs. 72.2%, respectively) antibodies in patients with HPV-positive OPC than in those with HPV-negative OPC [50]. In addition, they showed that HRs for both overall and progression-free survival among HPV-positive OPC patients were 0.2 for NE2, 0.3 for E1, and 0.3 for E6 antibody positivity. However, most reports have focused on E6 antibodies. As shown above, the diagnostic significance of E6 antibody elevation may be low due to positivity in nearly half of HPV-negative OPC patients [50].

Therefore, the association of pre- or post-treatment E6 antibody titer with the risk of recurrence has been investigated. With respect to pre-treatment serum E6 antibodies, conflicting reports have been published. Koslaova et al. and Fakhry et al. reported that being positive for E6 antibodies was associated with an increased risk of recurrence [33,51]. On the other hand, Lang Kuhs et al., Huang et al., and Dahlstrom et al. showed that positivity for E6 antibodies was associated with a decreased risk of locoregional recurrence and progression-free survival, respectively [50,52,53]. In addition, several studies reported no association between positive E6 antibody and recurrence [54,55]. With respect to post-treatment serum HPV E6 antibody levels, most reports have described no association with recurrence or progression-free survival [50,51,52,54,55].

Although significant elevations of serum IgG and IgA antibodies against EBV viral capsid antigen (VCA) have been observed in NPC patients, no significant change was noted between patients in continuous remission and those with tumor recurrence [11]. Similarly, serum HPV E6 antibody titers may not reflect tumor status after treatment.

## 7. HPV DNA in Oral Rinses of Recurrent Respiratory Papillomatosis Patients

Another well-known HPV-associated head and neck tumor is recurrent respiratory papillomatosis (RRP) [56]. RRP is recognized as a chronic disease, characterized by papillomatous growths in the airway, predominantly affecting the larynx and trachea. Typically, multiple exophytic lesions are observed in the larynx, resulting in voice disorders, as well as airway obstruction in cases with bulky lesions. Although high-risk types of HPV, predominantly HPV16, are detected in HPV-associated OPC, low-risk types of HPV, mostly HPV6 and HPV11, are detected in RRP. The disease is histologically benign. The age distribution of RRP in Europe is trimodal, with a peak in children at a median age of 7 years and two other peaks in adults at a median age of 35 and 64 years old [57]. Juvenile-onset RRP tends to have a more aggressive clinical course [58]. Therefore, due to repeated and frequent surgical procedures, the clinical impact on patients seems to be comparable to that of OPC.

A preliminary report by Born et al. showed that 25 (93%) of 27 adult-onset RRP patients were positive for HPV DNA in oral rinses [59]. Furthermore, Hao et al. recently reported that 22 (95.6%) of 23 RRP patients had an initial HPV-positive oral rinse [60]. They also showed that 17 (77.2%) of those 22 patients had an additional positive oral rinse in at least one of samples collected during 6–12 months following the acquisition of the initial rinse [60]. These results suggest that patients with RRP may have an increased propensity to harboring HPV in the oral cavity, as the prevalence of oral HPV infection in the general public is reported to be 6.9% (95% CI, 5.7%–8.3%) [59,61].

One of the interesting and important findings regarding RRP is that HPV DNA was detected in not only tumor tissue, but also in uninvolved sites in patients with active disease and in samples from patients in remission [62]. Therefore, the question of whether HPV DNA is detected in oral rinse samples from patients in remission is of great interest. Although Hao et al. analyzed multiple consecutive oral rinse samples from patients with RRP, the disease status at each time point of the oral rinse collection was not shown [60].

## 8. Future Perspective

There are several possible options available in regard to HPV testing using blood or oral rinse samples (Figure 2). The main risk factor of HPV-positive OPC is high sexual activity. D’Souza et al. stated that increasing the lifetime number of vaginal sex partners and oral sex partners resulted in a significantly high adjusted odds ratio for HPV-positive OPC [30]. HPV testing using blood or oral rinses could be a screening tool in cohorts with a high risk of HPV-associated OPC, including sex workers and patients in clinics for sexually transmitted infections. For this purpose, the improvement of sensitivity, as well as the development of a simple procedure to detect HPV, will be required. Because of the possible presentation of clinical relevance, the further development of a simple procedure to quantitate HPV DNA in plasma and oral rinses is also required.

Although it is known that HPV-associated OPC patients have a better prognosis than non-HPV-associated OPC patients, it is true that some HPV-associated OPC patients still show a poor prognosis. De-escalation of the treatment intensity for HPV-associated OPC has been vigorously investigated worldwide. However, de-escalated treatment should not be applied for patients with a poor prognosis. Based on the persistence of HPV-associated markers after treatment or the clearance of such markers, it may be appropriate to stratify treatment.

With respect to RRP, no data have been made available as to whether the detection of HPV in oral rinse samples has a significant clinical role. Morphologically apparent RRP can be easily observed through an endoscopy. However, as post-treatment HPV DNA in oral rinses predicts the prognosis of HPV-associated OPC, HPV DNA in oral rinses in patients in remission may predict the recurrence of RRP.

It is inevitable to discuss the various methods used to detect HPV DNA in plasma or oral rinse samples, and to measure the serum HPV titer obtained by various investigators. Recently, D’Souza et al. evaluated biomarker prevalence, and showed that oral rinse HPV testing had moderate-to poor sensitivity for HPV-associated OPC, whereas HPV E6 antibodies showed the best overall test performance [63]. In that study, the most sensitive method of HPV testing in oral rinse samples was HPV DNA detection using the DNA enzyme immuno-assay detection system, which showed 51% sensitivity and 99% specificity. As described above, the combined sensitivity of HPV DNA detection in oral rinses was much higher, and even the classic method of auto-nested PCR for HPV DNA, as used by Yoshida et al. and Murono et al., may be more useful [40,46,64]. Furthermore, in cervical premalignant diseases, droplet digital PCR (ddPCR) demonstrated a higher HPV DNA detection rate than that of quantitative PCR [65]. Larsson et al. also stated that ddPCR may provide a new promising tool for evaluating the HPV viral load in clinical samples [66]. Accordingly, Isaac et al. used ddPCR to detect HPV DNA in oral rinse samples, and demonstrated high sensitivity, as shown in Table 3 [41]. The establishment of a standardized method is desired and multi-institutional collaboration should be encouraged. Currently, no definitive conclusion can be drawn on this matter. However, further investigations could identify a way to accurately identify patients at risk of both the development and recurrence of HPV-associated OPC.

## Figures and Tables

**Figure 1 microorganisms-09-01150-f001:**
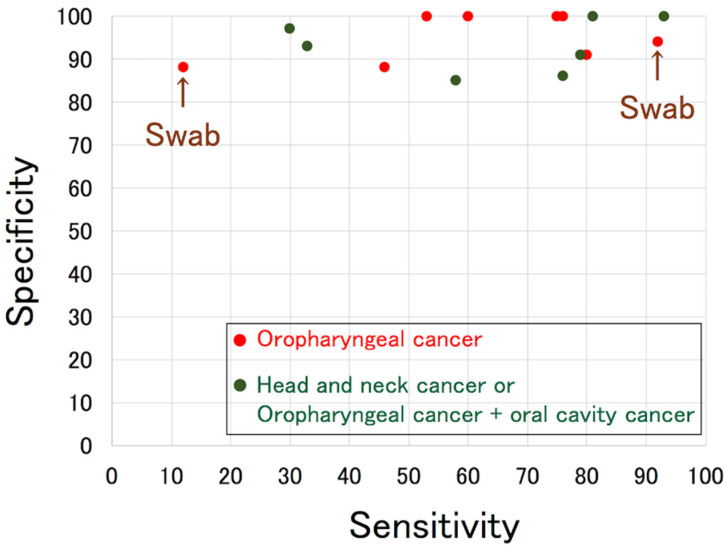
Scatter plot of sensitivity and specificity of pre-treatment HPV DNA detection in oral rinse in each study of oropharyngeal cancer. “Swab” indicates samples collected by swab.

**Figure 2 microorganisms-09-01150-f002:**
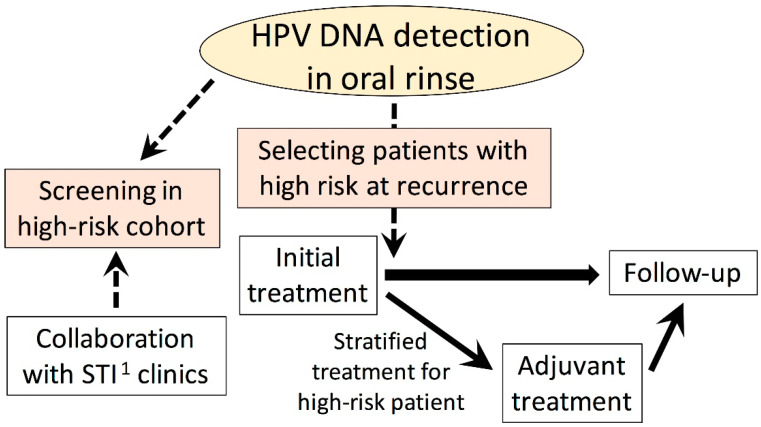
Future perspectives on utilizing HPV DNA detection in oral rinses in clinical settings. ^1^ STI, sexually transmitted infection. Modified from [47] with permission by the Society of Oto-Rhino-Laryngology, Tokyo.

**Table 1 microorganisms-09-01150-t001:** HPV DNA detection in pre- and/or post-treatment plasma in oropharyngeal cancer patients.

Author (Reference)	Year	Number of Cases (HPV-Positive: HPV-Negative)	Tissue HPV Status	Detection Method	Sensitivity ^1^	Specificity ^1^
Cao [20]	2012	HPV-positive OPC ^2^ 40 + HPV-negative HNC ^3^ 24	p16	qPCR ^4^	65%	100%
Ahn [21]	2014	OPC 87 (75:12) + unknown primary 6 (6:0)	ISH ^5^ or p16	qPCR	67% (pre-treatment) 55% (post-treatment)	100% (pre-treatment) 96% (post-treatment)
Dahlstrom [22]	2015	OPC 141 (114:27)	PCR	qPCR	61%	67%
Lee [23]	2017	(Test cohort) OPC 47 + LC ^6^ 4 + HPC ^7^ 4 (27:28) (Validation cohort) OPC 28 + LC 4 + HPC 1 (20:13)	p16	Amplicon-based next generation sequencing assay	100% 90%	93% 100%
Chera [24]	2019	OPC 103 (44:10:49 unknown) + HV ^8^ 103	p16	Digital PCR	89%	97%

^1^ Sensitivity and specificity of HPV DNA detection in plasma in regard to tumor tissue HPV status, ^2^ Oropharyngeal cancer, ^3^ Head and neck cancer, ^4^ Quantitative PCR, ^5^ In situ hybridization, ^6^ Laryngeal cancer, ^7^ Hypopharyngeal cancer, ^8^ Healthy volunteer.

**Table 2 microorganisms-09-01150-t002:** HPV DNA viral load in pre-treatment plasma in oropharyngeal cancer patients.

Author [Reference]	Year	Sample	HPV DNA Viral Load
Cao [20]	2012	plasma	<500 copies/mL in 13 patients and >500 copies/mL in 13 patients among 26 patients with detectable HPV DNA
Dahlstrom [22]	2015	plasma	0 copies/mL in 114 patients, 0.1 to <10 copies/mL in 23 patients, 10 to <100 copies/mL in 59 patients; 100 to <1000 copies/mL in 47 patients and ≥1000 copies/mL in 19 patients among all 262 OPC ^1^ patients, including 114 HPV-positive, 27 HPV-negative and 121 with patients missing data
Chera [24]	2019	plasma	Median 419 copies/mL, ranging from 8 to 22,579, in 84 HPV-positive OPC patients with detectable HPV16 DNA in plasma

^1^ Oropharyngeal cancer.

**Table 3 microorganisms-09-01150-t003:** HPV DNA detection in pre-treatment oral rinse in oropharyngeal cancer patients.

Author [Reference]	Year	Number of Cases (HPV-Positive: HPV-Negative)	Tissue HPV Status	Detection Method	Sensitivity ^2^	Specificity ^2^
Smith [29]	2004	OPC ^3^ 67 (25:42) + OCC ^4^ 126 (13:113)	PCR and direct sequencing	PCR and direct sequencing	58%	85%
D’Souza [30]	2007	OPC 100 (72:28)	ISH ^5^	PCR	32% (HPV16-positive rate)	NA ^6^
Gillison [31] *	2008	HNC ^7^ 240 (92:148) including OPC 114 (82:32)	ISH	qPCR ^8^	33%	93%
Agrawal [32]	2008	HNC 135 (44:91) including OPC 52 (41:11)	ISH	PCR + linear probe assay	30% (HPV16)	97% (HPV16)
Koslabova [33] *	2013	OPC 118 + OCC 24 (84:58)	PCR	PCR + line blot hybridization	76%	86%
Ahn [21] *	2014	OPC 87 (75:12) + UP ^9^ 6 (6:0)	qPCR	qPCR	53%	100%
D’Souza [34]	2014	HPV-positive OPC 164	ISH or p16	PCR and qPCR	61% (oncogenic HPV-positive rate)	NA
Dang [35]	2015	Mostly OPC 56 (48:8)	p16	qPCR	46%	88%
Rettig [36]	2015	HPV-positive OPC 124	ISH and p16	PCR + line blot hybridization	54% (HPV16-positive rate)	NA
Tsao [37] *,^1^	2016	OPC 144 (128:16)	ISH or PCR	PCR + Easy-Chip HPV blot	12%	88%
Chai [38] *	2016	HNC 82 (42:40) including OPC 50 (38:12)	p16 and ISH	qPCR	93%	100%
Imai [39]	2016	OPC 15 (5:10)	p16 and ISH	Cobas	60%	100%
Yoshida [40] *	2017	OPC 19 (12:7)	p16	Auto-nested PCR	75%	100%
Isaac [41] *,^1^	2017	OPC 52 (36:16)	p16	Droplet digital PCR	92%	94%
Rosenthal [42]	2017	OPC 45 + OCC 61 (43:63)	p16	Cobas	79%	91%
Wasserman [43]	2017	OPC 24 (17:7)	p16	Nested PCR	76%	100%
Tang [44]	2019	OPC 121 (89:32)	p16	qPCR	80%	91%
Fakhry [45]	2019	OPC 217 (187:30) + OCC 170 (7:163) + UP 9 (8:1)	mRNA	PCR + line blot hybridization	84%	88%

^1^ Samples were collected using swabs in these studies. ^2^ Sensitivity and specificity of HPV DNA detection in oral rinse samples in regard to tumor tissue HPV status, ^3^ Oropharyngeal cancer, ^4^ Oral cavity cancer, ^5^ In situ hybridization, ^6^ Not available, ^7^ Head and neck cancer, ^8^ Quantitative PCR, ^9^ Unknown primary, * References cited by combined sensitivity and specificity in [46]. Modifed from [47] with permission by the Society of Oto-Rhino-Laryngology, Tokyo.

**Table 4 microorganisms-09-01150-t004:** HPV DNA viral load in pre-treatment oral rinse in oropharyngeal cancer patients.

Author [Reference]	Year	Sample	HPV DNA Viral Load
Agrawal [32]	2008	Oral rinse	Median 4.6 copies/1000 cells, ranging from 0.2 to 19, in positive samples
Rettig [36]	2015	Oral rinse	Median 161 copies/2 µL, ranging from 21 to 846
Tang [44]	2019	Oral rinse	Mean 774.1 copies/50 ng in advanced stage of HPV-positive OPC ^1^ and 232.0 copies/50 ng in early stage of HPV-positive OPC

^1^ Oropharyngeal cancer.

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
