# Peer review of "Virus-Associated Biomarkers in Oropharyngeal and Nasopharyngeal Cancers and Recurrent Respiratory Papillomatosis"

_microorganisms, 2021, doi:10.3390/microorganisms9061150_

Round 1

Reviewer 1 Report

Well written and clear review.

Section 2.1 (Epstein-Barr virus) appears a bit abrupt in the manuscript and may be introduced in the last paragraph in the previous section.

Rewrite L60-61. «detectable» floats a bit in the sentence

L168 delete “it”

I am a bit sceptical to include Figure 2 as the review reads well without it. It is OK to report the preliminary data, but the Figure does not lift the paper. I would consider to publish a separate report on the data obtained. If so pursued, I think it is important to explain the choice of GP5+/GP6+ primers and what auto-nested PCR means. There is also the first lane is without explanation (I assume this is the remnants of the ladder). Having used specific primers for HPV6 and HPV11 might have given better results than the general primer set designed to cover a range of types, but should be discussed in all regards.

L249 add “in” so that it reads “also in unaffected…”

L256 rewrite to: “high sexual activity. D’Souza et al. stated that increasing the lifetime number of vaginal-sex…”

Make Figure 3 print-friendly and use consistent fonts

Reviewer 2 Report

The submitted manuscript reviewed the current progress in HPV DNA detection from oral rinse samples. After a brief history of viral DNA detection of EBV, it reviewed the HPV genome changes in oropharyngeal cancer before or post-treatment, serum HPV antibodies, and HPV DNA in RRP.

Major:

The submitted manuscript is titled Human papillomavirus detection in oral rinse of oropharyngeal cancer patients. However, many contents are not remotely related to "HPV," "oral rinse," and "cancer." For example:

  • The background section is irrelevant to the title.
  • Section 6 is about serum antibody.
  • RRP is not cancer.

Line 231/5, "Similarly, our unpublished data demonstrated that HPV DNA was detected in oral rinse in 5 (71%) of 7 patients with RRP (Figure 2). HPV typing was performed in 5 patients, and reveled HPV6 in four patients and unidentifiable in the remaining patient. Interestingly, HPV DNA was undetectable in tonsillar swab in all 7 patients (Figure 2)."

241/7, Figure 2:

  • Research data should not be published in a review. I personally object to publishing experimental data in a review. Since a review article does not include the Method section, such results are not reviewable nor repeatable by other researchers.

Minor:

Line 41/4, "Although cytology-based screening programs for cervical cancer were implemented in many countries, HPV testing was found to have higher sensitivity than cytology but somewhat lower specificity in cervical cancer. Therefore, co-testing with cytology and 43 HPV, and lately HPV testing alone, have been proposed for cervical cancer screening."

  • It is not clear what the term "HPV testing" is referring to. There are many HPV tests available. 
  • Please consider rewriting this sentence.

Reviewer 3 Report

The review manuscript entitled “Human papillomavirus detection in oral rinse of oropharyngeal cancer patients” by Dr. Shigeyuki Murono  underly the current knowledge behind the HPV DNA detection in plasma and oral rinse as well as serum anti-HPV 16 detection as HPV-associated oropharyngeal cancer diagnostic/prognostic markers.

Strengths

The manuscript is well written and concise.

Reference list is adequate.

Weakness

Despite the manuscript is well written and concise in general, the EBV section seems to be out of context compared to the focus of the manuscript

Major points

  1. It is unclear for the reviewer why two whole paragraphs are dedicated to Epstein-Barr virus and its employment in nasopharyngeal cancer as biomarker while these notions are not reported neither in the abstract nor in the title. Please revise the title and the abstract accordingly or, alternatively, strongly reduce/delete this paragraph.
  2. The manuscript mainly describe qualitative data from descriptive studies. What about the HPV DNA load determination? The quantitative determination of HPV DNA might presents clinical relevance.  
  3. The details of how specificity and sensitivity have been calculated throughout the mentioned studies should be included.

Minor revisions

Line 38 --> What about the novevalent vaccine?  For instance PMID: 31755549 MID: 31600572 PMID: 31063090

Lines 41-45--> As correctly indicated by the author, although the clear role of HPV in cervical carcinogenesis, the current standard screening test comprise the pap-smear test which is couplet with HPV-specific PCR Test. However, a more analytical assay recently emerged for facilitating the reduction of HPV-driven tumors, by increasing HPV detection rates is the HPV specific droplet-digital PCR (ddPCR) (PMID: 33329471; PMID: 28748500). Due to its high sensitivity in detecting HPV DNA, it can be applied for investigating clinical samples carrying a low amount of viral DNA, such as sera and/or oral rinses. For completeness, this information and references should be included.

Line 30-32 please revise the English

The original articles of the figures should be included.

Tables 1 and 2 please detail the meaning of specificity and sensitivity.

Round 2

Reviewer 2 Report

In my opinion, this review covered too many topics, and the organization of this review can be improved.

Reviewer 3 Report

The authors addressed all reviewer concerns and substantially improved the manuscript